# Spatiotemporal Decoupling of Population, Economy and Construction Land Changes in Hebei Province

**Mu Li [1,2], Yunyang Shi [1,2], Wenkai Duan [1,2,3] , Aiqi Chen [1,2], Nan Wang [1,2] and Jinmin Hao [1,2,*]**

[1] College of Land Science and Technology, China Agricultural University, Beijing 100193, China; ZGNDlm@cau.edu.cn (M.L.); yunyangshi@cau.edu.cn (Y.S.); dwk@cau.edu.cn (W.D.); aqchencau@126.com (A.C.); nwang@cau.edu.cn (N.W.)
[2] Center for Land Policy and Law, China Agricultural University, Beijing 100193, China
[3] Faculty of China Agricultural University Library, China Agricultural University, Beijing 100193, China
[*] Correspondence: jmhao@cau.edu.cn; Tel.: +86-010-6273-3568

**Abstract:** Under the context of rapid urbanization, how to use construction land resources under the dual pressure of socioeconomic growth and cultivated land protection is critical to resource utilization and sustainable development. Thus, it is of great theoretical and practical significance to study the relationship between socioeconomic change and construction land expansion. Based on decoupling theory, this study constructed a two-dimensional model to analyze the population-construction land and economy (non-agricultural GDP)-construction land decoupling status and characteristics in Hebei Province at the county level. Then, a decoupling-based construction land-use zoning model was built to explore construction land saving and intensive utilization strategies in different construction land-use zones. The results show that (1) the construction land area, population and non-agricultural GDP in Hebei Province increased in the study period, but there were spatial differences in the hot areas of growth. (2) In the population-construction land dimension, the growth of the population and construction land in Hebei Province was generally in an uncoordinated state. According to the results of the calculation, the samples of counties whose relationships between population and construction land were uncoordinated accounted for 75.76% of all counties, and 68.94% of all counties demonstrated weak decoupling. (3) In the economy-construction land dimension, 89.39% of all counties in Hebei Province had coordinated relationships between economy and construction land change. The expansion negative decoupling was the main decoupling state in Hebei Province in this dimension. (4) On the basis of two-dimensional decoupling type results, the construction land-use in Hebei was divided into four zones: "Population–economy dual coordinated", "population unilateral coordinated", "economy unilateral coordinated" and "population–economy dual uncoordinated". The results showed that the "economy unilateral coordinated" zone included 68.18% of all counties. According to the characteristics of different construction land-use zones, this study provided various regulatory and control countermeasures and suggestions to improve the efficiency of construction land-use and to promote sustainable development in Hebei Province.

**Keywords:** construction land; population; non-agricultural GDP; two-dimensional decoupling model; construction land-use zoning; Hebei Province

---

## 1. Introduction

Urbanization has become a global phenomenon in recent years, especially in China [1–3]. Since the implementation of the reform and opening-up policy in 1978, China has experienced dramatically rapid urbanization. Between 1978 and 2017, the demographic urbanization, which is the proportion of urban population to total population, increased from 17.92% to 58.52% [4–6]. With the development of

industrialization and urbanization, construction land (land for construction of buildings), which is an indispensable resource that supplies humans with spaces and elements for habitation and production, has expanded rapidly. Over the period of 2010–2017, the construction land in China increased by 39,079.46 km$^2$, with a total growth rate of 10.95% [7,8]. Meanwhile, the total population increased by 49.17 million [6], and GDP has continued to grow at a rate of approximately 7 to 9 percent per year [9]. It can be inferred that construction land has made an important contribution to the promotion of social and economic growth [10,11]. However, although the population, construction land and GDP have grown simultaneously, the problems of "shortage and waste" exist in the utilization of construction land. On the one hand, the expansion of construction land is at the expense of occupying a large amount of cultivated land [2]. During 2010–2017, the total reduction in cultivated land reached 29,478.70 km$^2$, of which approximately 80% was due to construction occupation [8]. To ensure national food security and protect the ecological environment, the Chinese government has implemented a strict cultivated land protection system. The system strictly controls the new construction land supply, resulting in a "shortage" of construction land resources. However, inefficient utilization of construction land also exists [12]. The per capita construction land area increased from 266.08 m$^2$ in 2010 to 284.78 m$^2$ in 2017. There is great potential for construction land consolidation in China. Therefore, there is a pressing need for China to utilize construction land efficiently and effectively [13]. For this reason, the Ministry of Natural Resources of China has released policies and measures such as "The management method for the pilot of increase vs. decrease of urban and rural construction land", "Regulations on saving and intensive use of land" and "Measures for the disposal of idle land" [14–16]. The purpose of these policies is to improve the efficiency of construction land-use by replacing the construction land index in different places, formulating the standard of construction land utilization and promoting the reuse and reutilization of stock construction land.

In recent years, the spatiotemporal change and distribution of construction land has been a hot topic. At present, the main problems of construction land-use in China are classified as "two-way expansion and double-sided inefficient". On the urban side, because the Chinese government relies firmly on land inputs [3,17,18], a large number of economic development zones have been established, in which land expansion is usually the first step. Because of the ignorance of local conditions or the inaccurate judgment of the economic situation, many well-established economic zones have not developed, leading to land-use inefficiency [3]. In addition, the potential of existing construction land in old urban areas has not yet been realized. As a result of the aging of the inner urban areas, the urban population moved to the outer areas of the new city, causing the urban construction land in the city to sit idle. In terms of rural areas, the rural population and construction land showed a reverse coordination trend, which means the rural construction land showed an increasing trend under the situation of the decreasing rural population [19]. There is a phenomenon of "village-hollowing", which is critical to rural construction land-use [20,21]. Some studies have shown that with increasing income, rural residents prefer to build new houses outside the village rather than reconstruct old ones [20,22]. Additionally, due to the lack of a withdrawal mechanism, the profit from removing construction land is very low, resulting in a low willingness to withdraw old houses. Consequently, the gradual expansion of the periphery and the "hollow" of the internal village both occurred [20–22]. Therefore, it is of great significance to analyze the relationships of population, economy and construction land changes, which is helpful for analyzing the rationality of construction land-use; additionally, this information helps to promote the efficient use of regional construction land and alleviate the contradiction between construction land and agricultural land.

The relationship between population or economy and construction land change has always been the focus of academic research. Many existing theoretical and practical studies have indicated that there is a significant interaction between population or economy and construction land change [23,24]. Wang et al. [25], Gao et al. [26] and Liu et al. [27] concluded that population change and economic growth were the main driving forces of construction land expansion. Furthermore, some scholars have analyzed the dynamic relationships of population-construction land (POPCL) and economy-construction land

(ECNCL) separately and in detail. For the POPCL relationship, Qi et al. analyzed the spatial distribution of daytime and nighttime populations and their influences on land-use differences in urban and rural areas in Haidian District [28]. de Espindola et al. studied the relationship between population change and land expansion in the urbanization process [29]. Referring to the research on the ECNCL relationship, Xie et al. studied the relationship between urban construction land expansion and economic growth of the Yangtze River economic belt [30]. Liu et al. proposed that the power function was more suitable than the linear function to describe the relationship between GDP and construction land area in the Pearl River Delta from 1979 to 2009 [31]. Ye et al. discovered the different circular pattern of the economic efficiency of construction land-use in the Pearl River Delta and proposed that the core part should focus on the reuse of inefficient development land, while the outer circle should pay attention to controlling the expansion of inefficient development land [32].

Among the many methods of analyzing the relationship between two factors, decoupling analysis is widely used. Decoupling analysis can be used to analyze the relationships between changes in two independent variables. This method aims to identify whether a change of one variable is coordinated with another variable change. It was first used by the OECD to evaluate the relationship between environmental pressure and economic growth [33]. At present, the decoupling method has been widely used in many fields, including its application in research on land-use, which has proven to be of good applicability. Jing W et al. and Chengcheng W et al. separately studied the relationship between population and construction land change in urban and rural areas from the national and provincial scales using the decoupling analysis method [18,19]. Du et al. evaluated the rationality of construction occupation of cultivated land according to the decoupling relationship between the growth of construction land and the decrease in cultivated land [34]. Zhong et al. analyzed the decoupling relationship between economic growth and construction land growth [35]. However, there are still some shortcomings in the existing research. Regarding the research scale, the national and regional scales have received much attention, leading to difficulties in providing reference for practice. As counties are the actual basic administrative units in China, in which policies, regulations and management measures are relatively unified, this paper takes the county-level units as the research unit. In addition, the production function of construction land is mainly reflected in non-agricultural production. Most current studies analyze the relationship between construction land and total GDP, which is inaccurate. Thus, this paper used the non-agricultural GDP (NAGDP) as the economic output to better reflect the utilization of construction land. From the perspective of the research methods, previous studies have focused on the relationship between construction land and other single elements, such as the relationships of POPCL or ECNCL, which are relatively unilateral. Studies involving both POPCL and ECNCL are relatively weak. Therefore, this paper combined the studies on the two-dimensional relationships of POPCL and ECNCL to attempt to reveal the main problems and shortcomings in the current utilization of construction land. Then, this paper proposed targeted countermeasures and suggestions, with an objective of improving the efficiency of construction land-use.

Regarding the choice of study area, we selected Hebei Province as the research object. In recent years, Hebei Province has experienced rapid socioeconomic development and urbanization as well as construction land expansion. However, as the main grain production area of China, Hebei Province bears the important responsibility for guaranteeing food security; thus, the protection of cultivated land is very urgent. The contradiction between construction land expansion and cultivated land conservation is becoming increasingly serious. How to make use of construction land under the dual pressure of economic development and cultivated land protection in Hebei Province is very important. It is necessary to conduct in-depth research on the relationship between construction land and socioeconomic change in Hebei Province to identify suitable construction land-use strategies. Therefore, it is of practical significance to choose Hebei Province as the research object.

In conclusion, this paper mainly studies (1) the spatiotemporal change in the construction land, population and NAGDP in Hebei Province; (2) the decoupling state of POPCL, which refers to the relationship between construction land and the population; (3) the decoupling state of ECNCL,

which refers to the relationship between construction land and NAGDP; and (4) the construction land-use zoning and corresponding suggestions for improving construction land-use efficiency in Hebei Province.

## 2. Materials and Methods

### 2.1. Study Area

Hebei Province is located in northern China, within 36°05′–42°40′ N and 113°27′–119°50′ E (Figure 1). It covers an area of 187,693 km², including various geomorphic types such as plain, hills, basin and plateau. Eleven prefectural-level cities and 168 county-level units are included in Hebei Province. Because of its unique location surrounding Beijing and adjacent to Tianjin, the development of Hebei Province is directly affected by these two municipalities. Since the implementation of the Integrated Development of the Beijing–Tianjin–Hebei Region Strategy, several industries have flowed to Hebei, resulting in rapid social and economic development. The GDP increased from 692.13 to 3401.63 billion yuan from 2003 to 2017. However, there are differences in the growth of the three industries: The tertiary industry has grown the fastest, followed by the secondary industry, while the primary industry has had the slowest growth (Figure 2a). Meanwhile, the total population has increased from 67.69 to 75.20 million over the same period, during which the urban population has increased faster than the rural population, resulting in an increase in urbanization (Figure 2b). As a result, the demand for construction land increased, which led to an increase in construction land area from 20,110.01 km² to 22,416.50 km² during 2009–2017. At the same time, the cultivated land decreased from 65,613.53 km² to 65,188.62 km², with a reduction of 424.91 km² in Hebei Province over the same period (Figure 2c). Conflicts among economic development, population growth and construction land expansion have become increasingly serious.

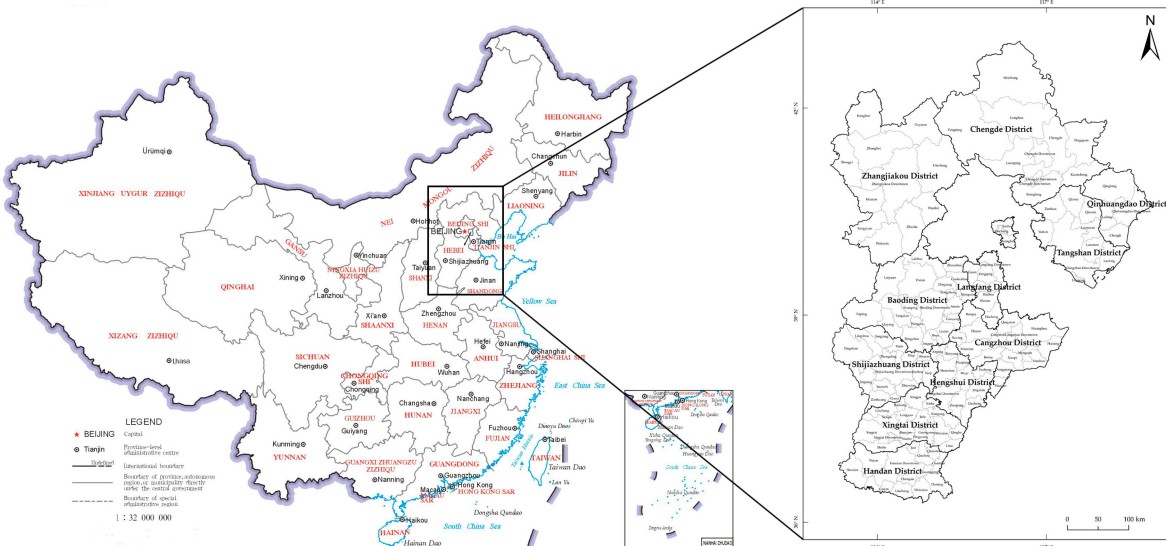

**Figure 1.** Location of Hebei Province in China. Note: This map of China is from the standard map system of the Ministry of Natural Resources of China.

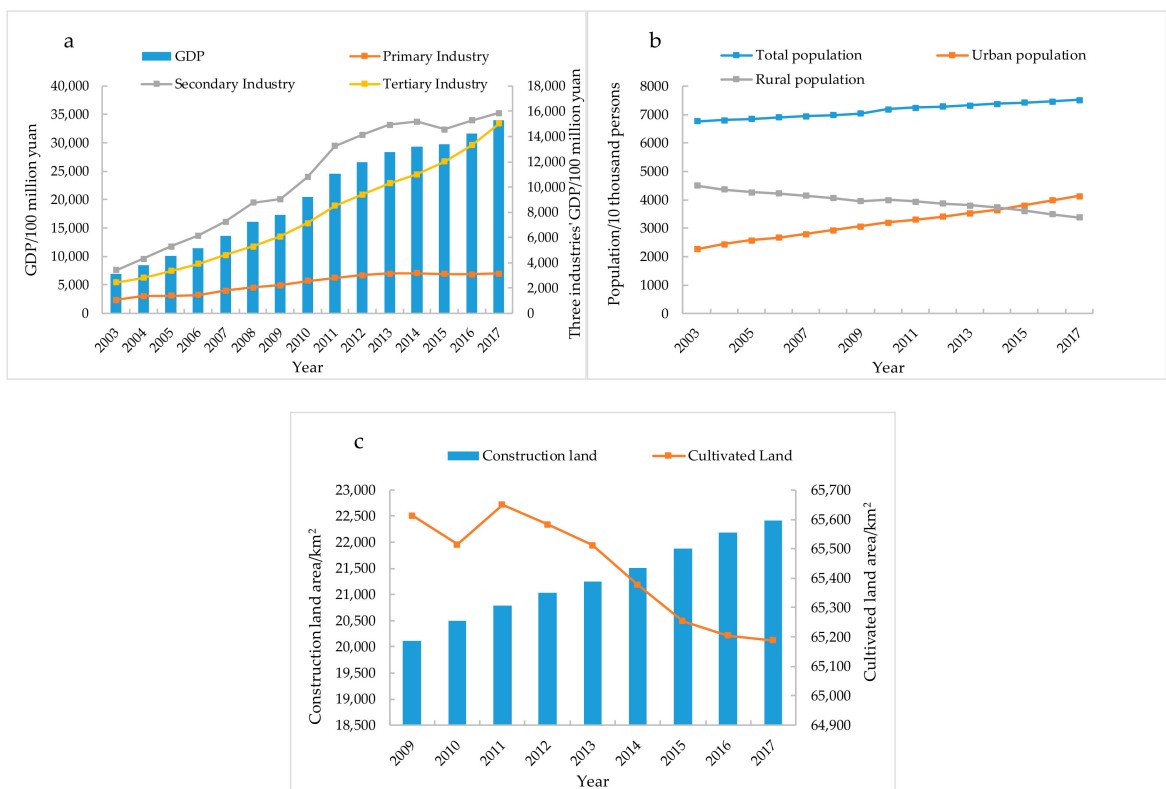

**Figure 2.** The changes in (**a**) the total GDP and three industries' GDP values; (**b**) the total population and urban, rural population; (**c**) the construction land farmland area.

### 2.2. Data Sources

The data of this study include population, construction land area, NAGDP and boundary data at the county level in Hebei Province. China has conducted the second national land survey and the subsequent land-use change survey since 2009, while the sixth population census of China was conducted in 2010. Therefore, considering the consistency of the data, this paper chose 2010–2017 as the study period. Due to data availability, counties in the downtown region of each city were merged correspondingly. Thus, this paper included 132 research units. Boundary information was obtained from the National Basic Geographic Information Center (http://ngcc.sbsm.gov.cn/). The demographic and economic data were derived from the Hebei Economic Yearbook (2011–2018). Construction land data were obtained from the Hebei Land Survey Statistical Yearbook (2010–2017). These data are a summary of the second land survey and subsequent land-use change survey in Hebei Province, and the data are more accurate. The construction land consists of three parts: Land for residential and industrial/mining sites, land for transport and land for water conservation facilities. The economic output data refer to the NAGDP data. To better explain the relationship between the economy and construction land area, the economic data were calculated at a constant price (2010 prices) by using the price indices.

### 2.3. Methods

#### 2.3.1. Two-Dimensional Decoupling Model

In this paper, a two-dimensional decoupling model centered on construction land-use was established, that is, the dimension of population-construction land decoupling and the dimension of economy-construction land decoupling. The "elasticity decoupling model" of Tapio was adopted to

analyze the decoupling relationship of population-construction land and economy-construction land, respectively [36]. The equation is as follows:

$$EC_i = \frac{\Delta X_i / X_{i0}}{\Delta L_i / L_{i0}} = \frac{(X_{it} - X_{i0}) / X_{i0}}{(L_{it} - L_{i0}) / L_{i0}} \quad i = 1, 2, 3, \ldots n,$$
(1)

where *ECi* is the elasticity value of construction and other factors in region *i*; $\Delta X_i$ is the change in population or NAGDP in region *i*; $\Delta Li$ is the change in construction land area in region *i*; $X_{it}$ and $X_{i0}$ are the population and NAGDP in the current year and base year, respectively, in region *i*; $L_{it}$ and $L_{i0}$ are the construction land area in the current year and base year, respectively, in region *i*.

According to Tapio's decoupling model, *EC* values of 0.8 and 1.2 are taken as the critical values for dividing the decoupling state. Eight states are divided as shown in Table 1 and Figure 3.

**Table 1.** Decoupling states of population-construction land and economy-construction land change.

| Decoupling State | ΔL | ΔX | EC | State Interpretation |
|---|---|---|---|---|
| Weak decoupling | ≥0 | ≥0 | 0 ≤ EC ≤ 0.8 | Both grow, construction land grows faster |
| Strong decoupling | >0 | <0 | EC < 0 | Construction land grows, population or economic output decrease; the worst state of construction land-use |
| Recessive decoupling | <0 | <0 | EC > 1.2 | Both decrease, construction land decreases slower |
| Expansive negative decoupling | >0 | >0 | EC > 1.2 | Both grow, construction land grows slower than population or economic output |
| Strong negative decoupling | <0 | >0 | EC < 0 | Construction land decreases, population or economic output increase; the best state of construction land-use |
| Weak negative decoupling | ≤0 | ≤0 | 0 ≤ EC ≤ 0.8 | Both decrease, construction land decreases faster than population or economic output |
| Expansive coupling | >0 | >0 | 0.8 ≤ EC ≤ 1.2 | Both grow and remain synchronized |
| Recessive coupling | <0 | <0 | 0.8 ≤ EC ≤ 1.2 | Both decrease and remain synchronized |

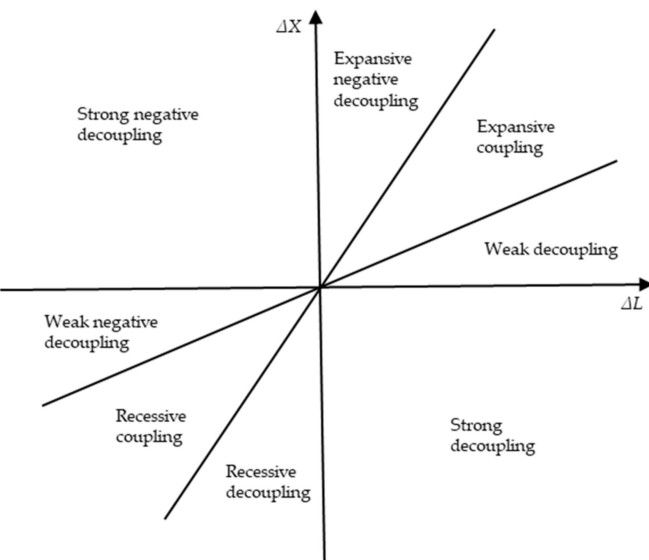

**Figure 3.** Diagram **of** decoupling states of population-construction land and economy-construction land change.

2.3.2. Framework of Construction Land-Use Zoning Based on the Two-dimensional Decoupling Model

This paper analyzed the decoupling states in two dimensions: (1) Decoupling state between construction land and population and (2) decoupling state between construction land and economy. Based on the two decoupling analyses, we can have two decoupling states of population-construction land and economy-construction land in each county. Then, 132 research units were divided into four

zones (Figure 4): (1) Population–economy dual coordinated zone, which means both dimensions (decoupling states of population-construction land and economy-construction land) show coordination; (2) population unilateral coordinated zone, which means the population-construction land dimension shows coordination, while the economy-construction land dimension is uncoordinated; (3) economy unilateral coordinated zone, where the economy-construction land dimension shows coordination, while the population-construction land dimension is uncoordinated; and (4) population–economy dual uncoordinated zone, which means both dimensions are uncoordinated. Among them, the construction land-use of the former three types are "coordinated" (two-dimensional dual coordinated or one-dimensional unilateral coordinated), and the fourth type of land-use is two-way uncoordinated.

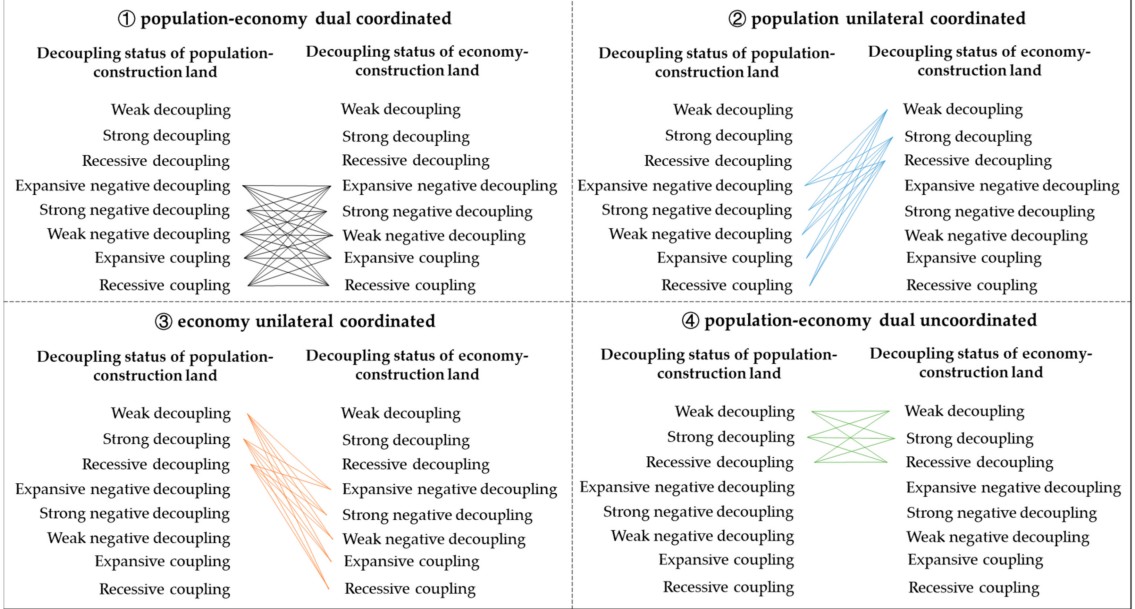

**Figure 4.** The relationship between construction land-use zoning and two-dimensional decoupling types.

### 2.3.3. Hot Spots Analysis Using Getis Ord *Gi*\* Statistic

Hot spots analysis is a widely used spatial analysis method. The Getis-Ord *Gi*\* statistic can be calculated for each feature in the data set. The features with either high or low values that are clustered spatially can be shown according to the z-scores and p-values [37].

$$G_i^* = \frac{\sum_{j=1}^t w_{i,j} x_j - \overline{X} \sum_{j=1}^t w_{i,j}}{S \sqrt{\left[t \sum_{j=1}^t w_{i,j}^2 - \left(\sum_{j=1}^t w_{i,j}\right)^2\right] / (t-1)}} \tag{2}$$

where $x_j$ is the value for feature $j$; $w_{i,j}$ is the spatial weight between feature $i$ and $j$; $t$ is the number of units, $t = 132$; $\overline{X}$ is the average value of $x$; S is the standard deviation; $G_i^*$ statistic is the z-score. A higher positive z-score shows more intense clustering of high values (hot spots), and a lower negative z-score represents more intense clusters of low values (cold spots) [38–40]. This paper used the natural break method of ArcGIS to divide the z-score into four types: Cold spots, cool spots, warm spots and hot spots. This paper used this method to analyze the spatial agglomeration characteristics of growth amount and rate of population, economy and construction land between 2010 and 2017. Hot spots represented the clusters of high values of growth amount and rate of three variables. Conversely, low spots represented the clusters of low values of growth amount and rate of three variables.

## 3. Results

### 3.1. Changes in Construction Land, Population and Economy in Hebei Province

#### 3.1.1. Changes in Construction Land in Hebei Province

As shown in Table 2, the construction land area showed a growth trend during 2010–2017 in Hebei. The area of land for residential and industrial/mining sites increased the fastest, with a growth amount of 1565.50 km$^2$, accounting for 81.3% of the increase in total construction land area. In addition, the land for transport increased by 331.89 km$^2$; however, its total growth rate was as high as 20.54%. The area of land for water conservation facilities had the slowest growth. From 2010 to 2017, the structure of construction land changed significantly. The proportions of land for residential and industrial/mining sites and land for water conservation facilities both decreased. However, the proportion of land for transport increased gradually and was mainly affected by the national substantial promotion of infrastructure construction. As a connecting area between Beijing and other provinces, Hebei Province plays an important role in the implementation of integrated development of Beijing, Tianjin and Hebei. Thus, the improvement in its regional connectivity through the construction of infrastructure such as highways and high-speed railways has been important work in Hebei.

**Table 2.** The changes in the population in Hebei Province during 2010–2017 (unit: km$^2$).

| | 2010 | | 2017 | | 2010–2017 | |
|---|---|---|---|---|---|---|
| | Area | Proportion (%) | Area | Proportion (%) | Growth | Growth Rate (%) |
| Total Construction Land | 20,490.89 | 100 | 22,416.50 | 100 | 1925.61 | 9.40 |
| Land for Residential, Industrial/Mining Sites | 17,814.31 | 86.94 | 19,379.81 | 86.45 | 1565.50 | 8.79 |
| Land for Transport | 1616.04 | 7.89 | 1947.93 | 8.69 | 331.89 | 20.54 |
| Land for Water Conservation Facilities | 1060.54 | 5.17 | 1088.76 | 4.86 | 28.22 | 2.66 |

For further study of spatiotemporal changes in construction land area, we performed a spatial analysis on the growth of construction land using the hot spots analysis tools in ArcGIS. Figure 5 presents the results of the growth amount (a) and rate (b) of hot spots of construction land. The results showed that during 2010–2017, the hot spots of construction land growth were mainly concentrated in the downtown region of Tangshan and its six surrounding counties, while the cold spots were mainly distributed in the middle and southeast of Hebei (Figure 5a). For the growth rate of construction land, the hot spots were mainly concentrated in the downtown area of Chengde and its three surrounding counties, the east of Langfang and Xingtai (Figure 5b). Moreover, it was apparent that along the line of southeast of Baoding-northwest of Hengshui-southeast of Hengshui, the growth amount and rate showed a "dual low" pattern, which indicated that the growth of construction land in these regions was the slowest in Hebei Province.

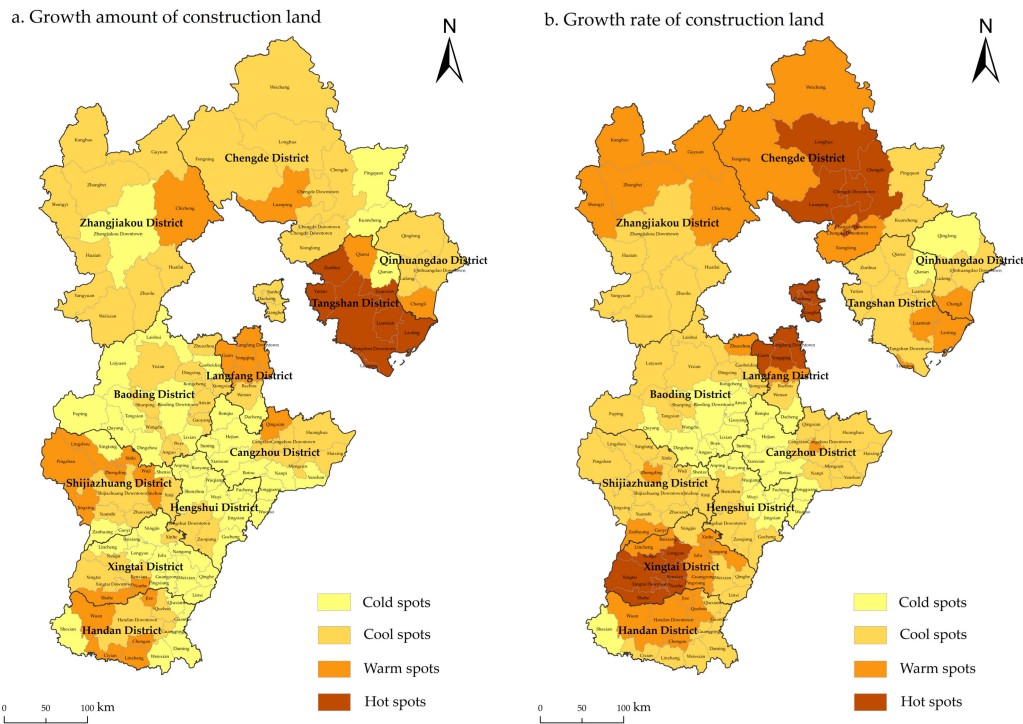

**Figure 5.** Hot spots analysis on expansion amount (**a**) and rate (**b**) of construction land area.

### 3.1.2. Changes in Population in Hebei Province

Figure 6 reveals that the total population of Hebei Province showed a growth trend during 2010–2017. The total population increased by 3.26 million, with a growth rate of 4.53%. Additionally, the urban population growth rate was 29.22%, while the rural population reduction rate was 15.26%. Thus, the proportion of urban population increased from 44.50% to 55.01% during 2010–2017. In 2015, the proportion of the urban population exceeded that of the rural population for the first time. The demographic gap between urban and rural areas gradually expanded, which was mainly caused by rural migration to urban areas.

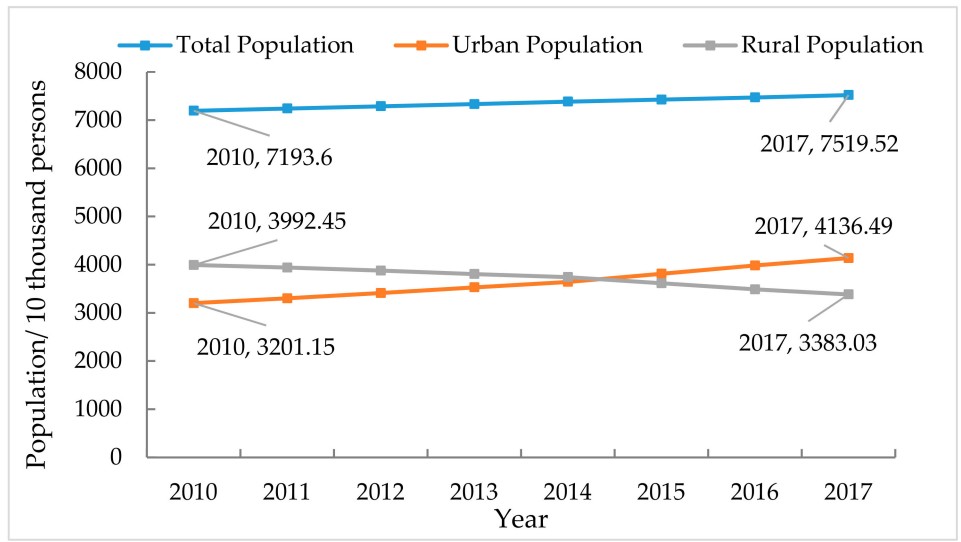

**Figure 6.** The changes in the total population and the urban and rural population during 2010–2017.

The population change was analyzed using the hot spots analysis tools of ArcGIS, the results of which are shown in Figure 7. As shown in the figure, in the downtown area of Shijiazhuang and its

surrounding nine counties as well as Yutian County in Tangshan were the hot spots in terms of the amount of population growth (Figure 7a). In terms of the growth rate, the areas of high growth rate have been along the line of the downtown of Cangzhou–the downtown of Langfang–Sanhe County. It can be assumed that the concentrated areas of demographic migration of Hebei were mainly in Shijiazhuang, Langfang, Cangzhou and Tangshan. These districts are the capital of Hebei Province (Shijiazhuang), the developed area (Tangshan) and the areas close to Beijing (Langfang) or Tianjin (Cangzhou), which easily attract population flow.

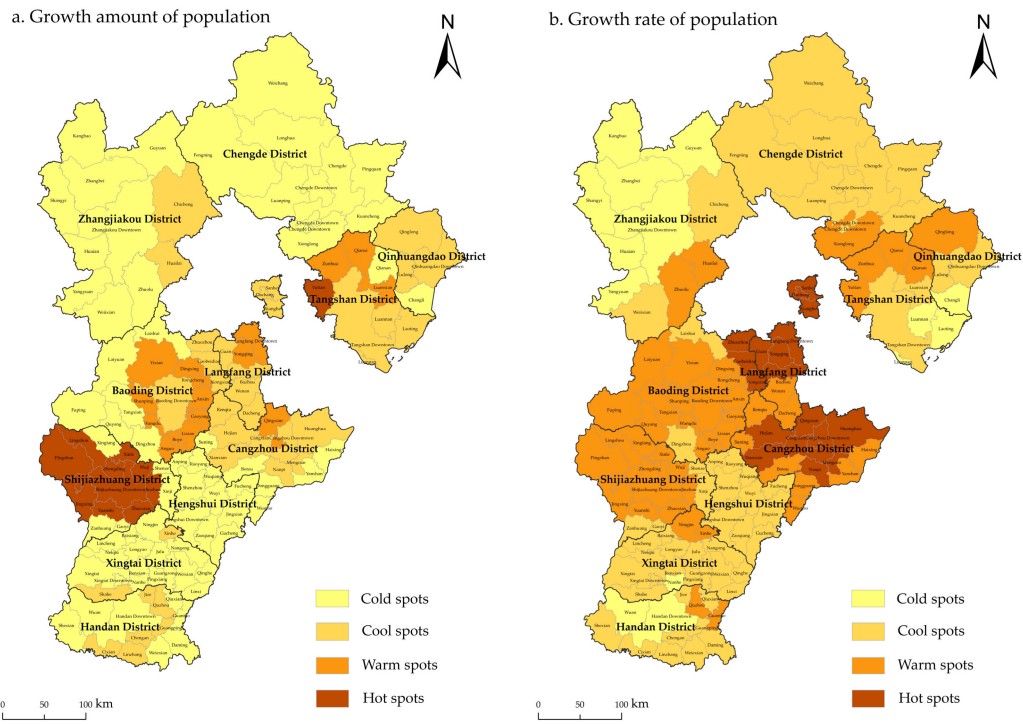

**Figure 7.** Hot spots analysis of growth amount (**a**) and rate (**b**) of population.

### 3.1.3. Changes in the Economy (NAGDP) of Hebei Province

As shown in Figure 8, during 2010–2017, the NAGDP of Hebei Province increased progressively. In the study period, the NAGDP increased with an average annual growth rate of 6.51%. In terms of the inner structure, the proportion of the secondary industry's GDP has shown a significant downward trend, while the tertiary industry's GDP has significantly increased, accounting for 48.7% of the total GDP in 2017. Furthermore, the gap between the secondary and tertiary industries' GDPs has continuously narrowed. It could be deduced that the proportion of the tertiary industry's GDP will exceed that of the secondary industry's GDP in the next two years under the current development trends. Specifically, the proportion of the tertiary industry's GDP has increased by 2.24% per year since 2015 and was obviously higher than that between 2010 and 2014 (0.53% per year). This result indicated that since the release of "The Outline of Collaborative Development of Beijing, Tianjin and Hebei" in 2015, Hebei has considered industrial upgrading as one of the important development strategic goals. By promoting the "Innovation and Entrepreneurship" policy, which aimed to promote high-quality development and industrial upgrading continuously, Hebei has made some achievements.

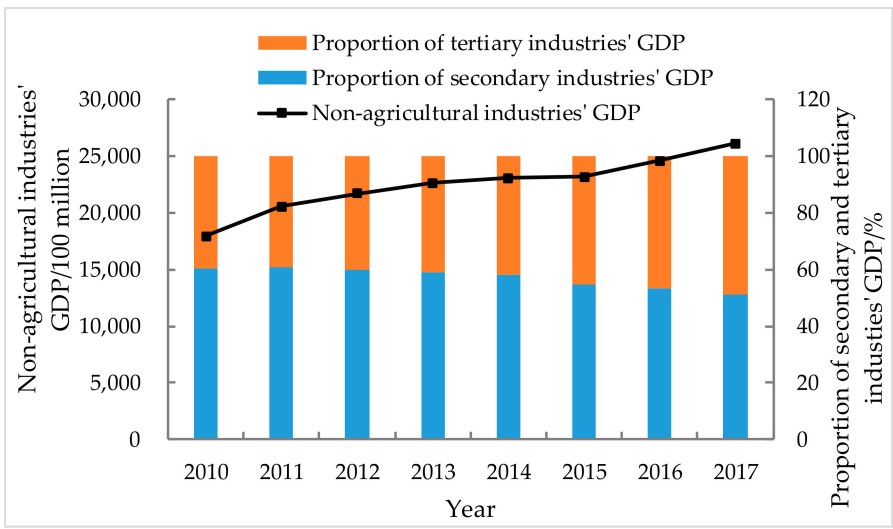

**Figure 8.** Changes in the non-agricultural GDP (NAGDP) and its components during 2010–2017.

Figure 9 shows the hot spots of economic growth, i.e., the NAGDP growth, by using the hot spots analysis tools of ArcGIS; Figure 9a,b presents the growth amount and rate, respectively. The results showed that the hot spots of the amount of economic growth were mainly distributed in Langfang and along the line from Shijiazhuang to Hengshui. However, the hots pots of the rate of economic growth were mainly concentrated in Handan–Hengshui, north of Langfang-northeast of Baoding and counties in eastern Zhangjiakou. Overall, the economic growth of Hebei Province exhibited an obvious distribution trend of "high in the south and low in the north" and formed a multi-core spatial pattern of economic development. Specifically, Shijiazhuang, as the capital of Hebei Province, has had strong economic growth momentum. In addition, due to the unique location close to Beijing, Langfang has obviously benefited from strong economic spillover from Beijing. Consequently, economic development in Langfang has been relatively fast, with its economic growth amount and growth rate showing a "dual high" pattern.

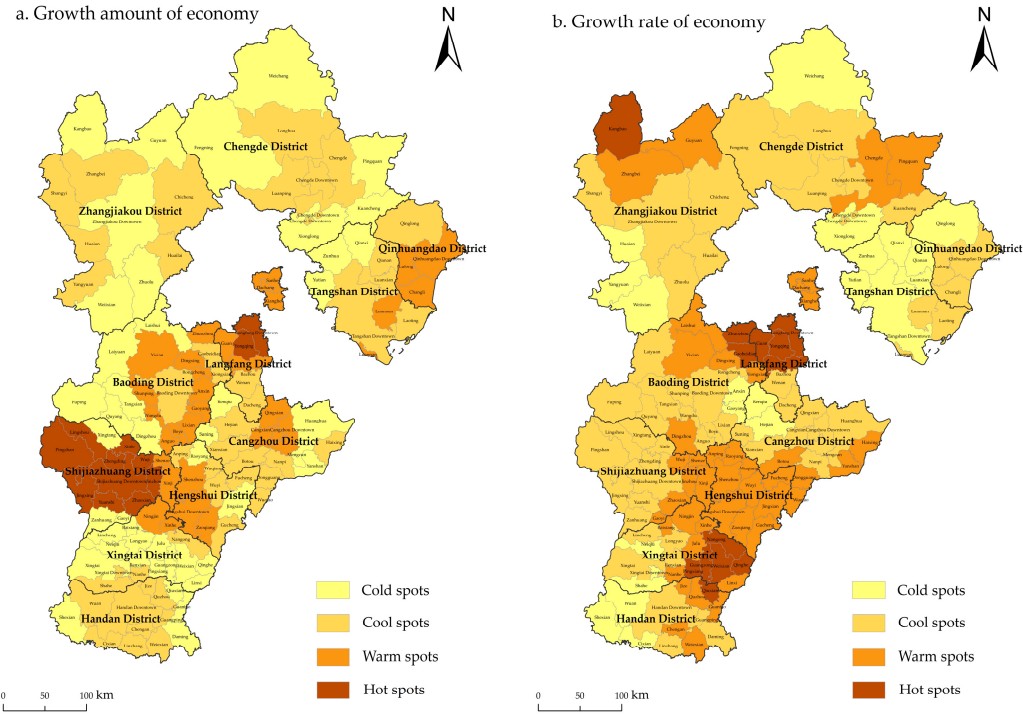

**Figure 9.** Hot spots analysis of growth amount (**a**) and rate (**b**) of NAGDP.

### 3.2. Two-Dimensional Decoupling Status of POPCL and ECNCL

#### 3.2.1. Decoupling Status Between Population and Construction Land (POPCL)

The relationship between the total population and construction land during 2010–2017 in Hebei could be divided into four types: Expansive negative decoupling, expansive coupling, weak decoupling and strong decoupling (Figures 10a and 11a). The POPCL relationships in most counties have been in an uncoordinated state. Ninety-one counties showed weak decoupling; that is, although population and construction land both increased, the growth rate of the total population was slower than that of the construction land in these counties. In addition, nine counties had strong decoupling, which was the worst POPCL relationship, where population decreased while construction land increased, and these counties were mainly distributed in northwestern Zhangjiakou and at the junction of Tangshan and Qinhuangdao. In contrast, there were 32 counties whose POPCL relationship was in a coordinated state. Among them, 20 counties in middle, eastern and southern Hebei (downtown of Shijiazhuang, junction of Tangshan and Qinhuangdao, Yi county–Anxin county–downtown of Cangzhou) exhibited an expansive coupling state, which was the best state of the POPCL relationship, indicating that population and construction land showed a trend of coordinated growth. There were 12 counties in Cangzhou, Tangshan and Zhangjiakou with expansive negative decoupling, which means that the growth rate of the construction land was slower than that of the population, indicating that the POPCL relationship in these counties was also relatively coordinated.

#### 3.2.2. Decoupling Status Between Construction Land and Economy (ECNCL)

For the relationship between the economy and construction land, the categories during this period were as follows: Expansive negative decoupling, expansive coupling, weak decoupling and strong decoupling (Figures 10b and 11b). In general, the ECNCL relationship in Hebei at the county level showed a coordinated trend. The samples with expansive negative decoupling accounted for 88.64% of all counties, whose economic growth rates were faster than those of construction land. Only one county was in an expansive coupling state. Additionally, there were seven counties with strong decoupling; they were mainly distributed in Baoding, Chengde, Handan and Xingtai, places in which the economy decreased while construction land increased. It is of great importance to control the construction land in these counties. Seven counties in Handan, Xingtai and Tangshan showed weak decoupling, where the growth rate of construction land was slower than that of the economy.

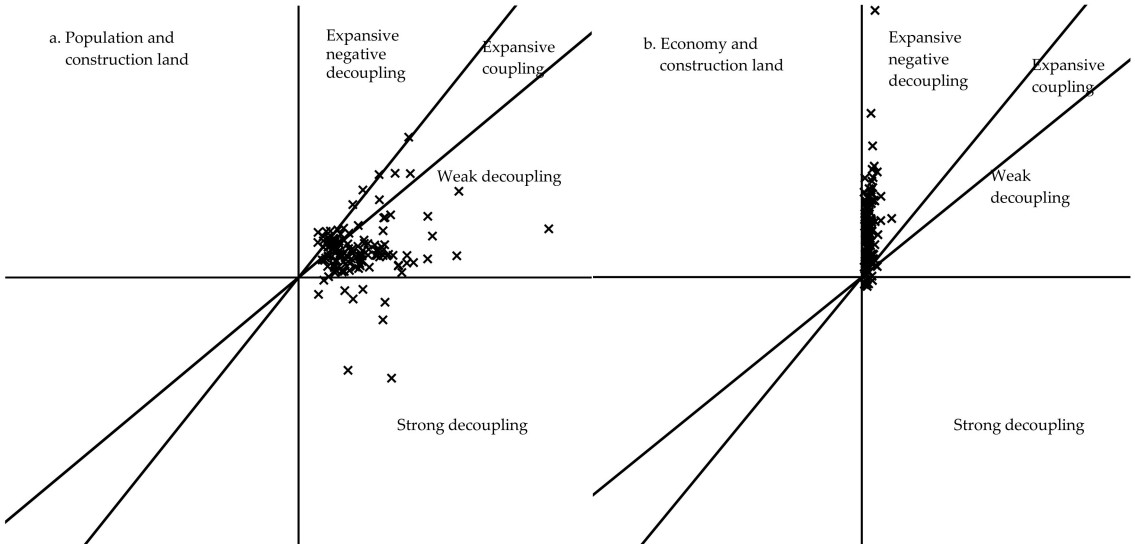

**Figure 10.** Coordinate distribution of decoupling state of population-construction land (POPCL) (**a**) and economy-construction land (ECNCL) (**b**) during 2010–2017 in Hebei.

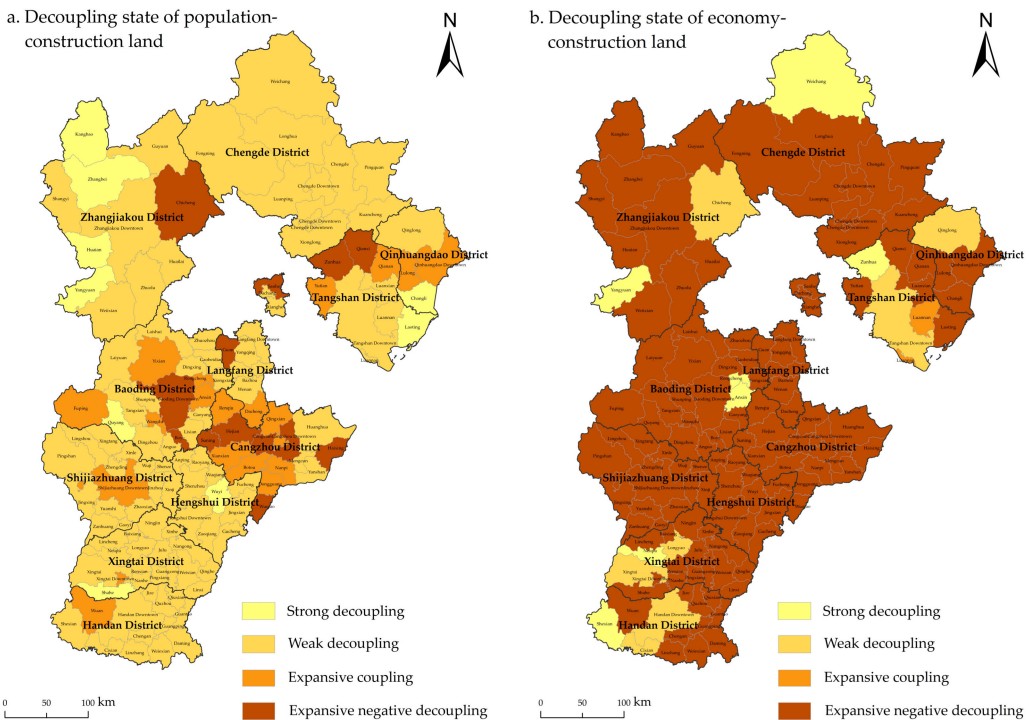

**Figure 11.** The spatial distribution of decoupling state of POPCL (**a**) and ECNCL (**b**) between 2010 and 2017 in Hebei.

### 3.3. Construction Land-Use Zoning and Corresponding Strategies in Hebei Province

In light of the two-dimensional decoupling analysis results above, based on the framework in Figure 4, which shows the relationship between construction land-use zoning and two-dimensional decoupling types, this paper divided the counties of Hebei into four construction land-use zones: Population–economy dual coordinated, population unilateral coordinated, economy unilateral coordinated and population–economy dual uncoordinated. Figure 12 exhibits the spatial distribution of the four construction land-use zones. It is apparent from the figure that the counties in the economy unilateral coordinated zone were the most widely distributed, accounting for 68.18% of all counties, followed by the counties with the population–economy dual coordinated type, with a proportion of 21.21% to all counties. Furthermore, corresponding suggestions for each zone were proposed.

(1) Population–economy dual coordinated zone: Both relationships of POPCL and ECNCL change showed a coordinated state. There were 28 counties in this zone, accounting for 21.21% of all counties. Spatially, counties in this zone were mainly distributed in the Shijiazhuang downtown and Xingtai downtown areas, along the line of Cangzhou–Baoding and the line of Qinhuangdao downtown–Qian'an County–Yutian County. As the capital of Hebei Province, Shijiazhuang has a strong attraction to population and economy. During the study period, the population and NAGDP increased by 503.2 thousand people and 102.42 billion yuan, respectively, and ranked first place in the increase in Hebei Province. In addition, in recent years, under the implementation of "saving and intensive use of land", the growth of construction land in Shijiazhuang has been stable, resulting in the coordinated state of growth among population, economy and construction land. In addition, Langfang, because of its unique location, has been significantly influenced by the radiation of Beijing, leading to a great attraction to the population and economy itself. Specifically, many counties in Cangzhou were in this zone, mainly because the population, economy and construction land all increased at relatively low levels, resulting in the two-dimensional coordinated state. Thus, it should be noted that although in the same construction land-use zone, different policies and measures should be taken in response to different situations, such as the level of the city, the stage of development and the strategy of development. On the one hand, for downtown Shijiazhuang, downtown Baoding, Xianghe County and

Gu'an County in Langfang, where population and economic would continuously grow in the future, the proper measures should relax the approval of construction land appropriately. With maintaining the coordinated development, the new construction land index should be increased appropriately. On the other hand, for counties in Cangzhou, promoting the development of economy and attracting population should be given more attention and the efficiency of construction land can be improved to realize high-efficiency and coordinated development in these regions.

(2) Population unilateral coordinated zone: The POPCL relationship was coordinated, while the ECNCL relationship was uncoordinated. There were four counties in this zone; they were distributed in northwestern Tangshan (Zunhua County), eastern Zhangjiakou (Chicheng County) and eastern Baoding (Anxin County and Rongcheng County). Among them, the POPCL relationship showed a state of expansive coupling in Anxin County and Rongcheng County and those in the other two counties showed expansive negative decoupling. The growth between population and construction land both showed coordinated and intensive trends. In contrast, the ECNCL relationship exhibited strong decoupling in Anxin County, Zunhua County and Rongcheng County, which meant that the NAGDP decreased while the construction land increased. In addition, the ECNCL relationship in Chicheng County displayed weak decoupling, which meant that the growth rate of NAGDP was lower than that of construction land. All four counties showed an uncoordinated state of ECNCL, which indicated that economic interest was the main factor that restricted the efficiency of construction land-use locally. Therefore, the main goal of construction land-use in this zone in the future should be the use of certain measures to control the new construction land index, to redevelop the inefficient or idle industrial land and to improve the economic intensity of construction land-use. Local governments should carry out the "dual control" measures proposed in China's 13th five-year plan, which emphasized that the "quantity" and "intensity" should receive equal attention in the period after construction land-use. Additionally, the access threshold of construction land-use should be improved by formulating industrial land-use standards and other measures to reverse-force the improvement of the economic efficiency of construction land-use.

(3) Economy unilateral coordinated zone: The ECNCL relationship was coordinated, while the POPCL relationship was uncoordinated. The counties in this zone were the most widely distributed in Hebei Province and included 90 counties, accounting for 68.18% of all counties. This finding indicated that the growth of economy and construction land in most counties tended to be coordinated, while that of the population and construction land trend were not. Spatially, these counties were widely distributed in Zhangjiakou, Chengde, Baoding, Hengshui, Shijiazhuang and eastern Handan. Among them, most counties showed a state of expansive negative decoupling in the ECNCL relationship, which means that the growth of NAGDP was much faster than that of construction land. In terms of the decoupling relationship of POPCL, eight counties displayed strong decoupling, in which the population decreased while the construction land increased. In addition, the other 82 counties showed weak decoupling. In contrast to the population unilateral coordinated zone above, counties in this zone faced the main problem of the population restricting construction land-use efficiency compared to economic benefits. Therefore, to optimize the construction land-use in these counties, a good living environment and better connection with other regions should be established and strengthened, resulting in the attraction of population and labor forces. Construction land strategies should shift and turn the quantitative advantage into the quality advantage. In addition, more attention should be paid to the structure of construction land supply, such as reducing the supply of residential land appropriately and improving the supply of industrial and commercial land, land for public infrastructure and urban green space.

(4) Population–economy dual uncoordinated zone: The changes in the relationships of both POPCL and ECNCL showed an uncoordinated state. There were 10 counties included in this zone, which were mainly scattered in Handan, Xingtai and Chengde (Figure 12). In most counties, the POPCL and ECNCL relationships both showed weak decoupling, which meant that both population and NAGDP growth were slower than construction land expansion. From two dimensions, the construction

land-use in these counties in this zone showed an uncoordinated state. Thus, the problems faced by these counties were from two sides: The population and economy. In the future, to improve the efficiency and intensity of construction land-use in these counties, the methods of rational expansion and reuse of idle land should be taken simultaneously by adopting multiple policies and measures. First, the allocation of new construction land indicators should be strictly controlled in land-use planning. In addition, through the demarcation of "Three Lines", which are the permanent basic farmland protection red line, urban growth boundary and ecological protection red line, the construction land could be forced into intensive use. Moreover, the local government should formulate construction land-use standards and pay attention to the supervision after implementation to improve the threshold of construction land-use. Last, stock construction land will be a treasure in the future; therefore, the implementation of measures such as land consolidation, land reclamation and land redevelopment and utilization should be encouraged to invigorate the stock construction land.

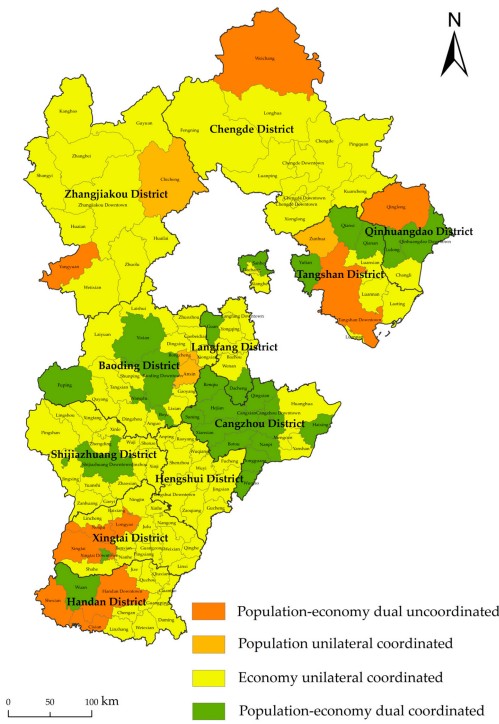

**Figure 12.** Spatial distribution of construction land-use zoning in Hebei at the county level based on the two-dimensional decoupling model.

## 4. Discussion

Construction land is an important land-use type that supplies humans with spaces and elements for habitation and production and the process of transforming other land-use types to construction land is considered irreversible [3,18]. Therefore, population and economy are important indicators that can be used to evaluate the utilization of construction land. The two-dimensional decoupling model framework proposed in this paper helps us understand the POPCL and ECNCL relationships in Hebei Province. From the current results, there were mismatches between the allocation of construction land and the other two indices, and the POPCL relationship was relatively poor. At present, the supply sources of construction land in China are mainly from new construction land and stock construction land. Generally, the new construction land has received much attention, while the reuse of stock construction land has been ignored, which will undoubtedly lead to the waste of construction land. With regard to the "zero growth planning" and "reduction planning", a multisource, dynamic, flexible mechanism of construction land allocation is needed, which calls for the "two-track" management

system: Both allocating the new construction land efficiently and reinvigorating the stock construction land at the same time.

### 4.1. Spatiotemporal Dynamic and Flexible Mechanism of New Construction Land Index Allocation

Influenced by the planned economy system, the new construction land in China is mainly allocated through the index management system. The new construction land index is determined in the Land-use Master Planning guide, whose temporal scale tends to be 10–15 years [41]. Because of information asymmetry and limited rationality of decision makers, it is difficult to predict economic and social development in the next 15 and 20 years [42]. In addition, the overall and annual index is allocated by the central government with the incremental construction land quota system, which is "top-down" [41,43]. Because of the low cost of irrational use of construction land after acquisition, there is often a phenomenon of "difficult to obtain and easy to use". This pattern may lead to the local government neglecting the actual need and constantly increasing the demand for construction land to obtain more construction land indicators from governments at higher levels. It can be inferred that the index of construction land lacks flexibility and rationality [42]. It is difficult to keep construction land change coordinated with socioeconomic development. Consequently, there is a phenomenon in which construction land-use is insufficient in some regions, while it is in a surplus in other regions. In addition, index exchange or transfer between different regions is not allowed [44]. Therefore, the establishment of the new index of "Marketization" under the framework of the current construction land allocation index system is of great practical significance [41,44,45]. In response to these two problems, some innovative thoughts and measures have been launched theoretically and practically. Under the existing construction land index allocation systems, scholars and local governments have proposed the market mechanism of the construction land index based on the land development rights theory. This approach, by means of market mechanisms, helps allocate the new construction land index among regions with less political dispute, leading to the coordinated result of socioeconomic development growth and construction land change.

### 4.2. From "Incremental-Based" Planning to "Stock-Based" Planning

Land-use planning in China is mainly based on incremental planning. Due to the lower cost of construction land expansion to new areas than that of the reconstruction of old areas, the governments tend to expand the urban areas outside instead of inside, leading to a higher likelihood of calculating incremental construction land indices. However, contrary to the strict control of the new construction land index, the management following utilization after index acquisition is insufficient, and results in the low efficiency of construction land. With the development of the socio-economy and urbanization, the "stock" construction land accumulates over time and is a wasted resource. Chinese governments have realized the importance of reusing stock construction land in both urban and rural areas. In urban areas, stock construction land is mainly low-efficiency and abandoned industrial land. With the upgrading of the industrial structure, there are a large number of restrictions, eliminations and prohibitions of industrial land in urban construction areas, which could be a main resource for future development [46], while in rural areas, as a result of the rural population immigration outside, the "hollowing village" [20] has been considered the main object to address. To promote the reuse and redevelopment of stock construction land in urban and rural areas, the Chinese government implemented a series of regulations and policies, such as "guidance on promoting conservative and intensive use of land", "measures for the disposal of idle land" and "the management method for the pilot of increase vs. decrease of urban and rural construction land" [14–16]. Local governments such as Shanghai, Chongqing and Guangzhou also explored the reuse and redevelopment of low-efficiency industrial land and old urban–rural areas in a series of innovative experiments that can be referenced by other regions [46–49]. Analyzing the relationship between construction land expansion and population, economic change could be used to help identify problems in construction land utilization, which serve as a good theoretical reference for "stock-based" planning.

*4.3. Priorities in Future Research*

Based on the two-dimensional decoupling model, this paper analyzed the POPCL and ECNCL relationships. However, improvements can still be made in future works. First, due to the limitation of the data, this paper studied only the relationship among construction land, population and economy in a short time series. The evolutionary characteristics and laws of relationships will be more obvious with the use of long-term data. Therefore, long-term data should be added in future studies. To improve the availability of data, remote sensing and statistics could be used to provide the database and guarantee further research. Second, the construction land-use zoning in this paper is divided according to the decoupling status based on the two-dimensional decoupling model. However, the decoupling status is based on the elasticity coefficient, which is the ratio of the growth rates. Thus, the present state cannot be shown using this method. In further works, the method of combining the decoupling type and the present state needs to be developed. Third, the research on the driving mechanism of the relationships will also be a focus of further research to explore the point of focus on the coordination and development of the relationship among construction land, population and economy.

## 5. Conclusions

This paper analyzed changes in construction land area, population and NAGDP. In addition, based on the decoupling theory, this paper analyzed the two-dimensional decoupling status of "population-construction land" and "economy-construction land" between 2010 and 2017. Construction land-use zones have been divided, and corresponding regulations and control countermeasures and suggestions have been proposed. We came to the following conclusions: The construction land area, population and NAGDP of Hebei Province all increased in the study period. However, there were spatial differences in the growth hot spots. The analysis of two-dimensional decoupling status indicated that the growth of economy and construction land in most counties tended to be coordinated, while that of the population and construction land trend were uncoordinated. In the population-construction land dimension, the growth of the population and construction land in most counties (75.76%) in Hebei Province were in an uncoordinated state. In the economy-construction land dimension, the growth of economy and construction land showed uncoordinated in most counties (89.39%). The construction land-use in Hebei Province was divided into four zones, and it was dominated by the "economy unilateral coordinated" zone, which had 68.18% of all counties.

This paper attempted to make some progress in the study method on construction land-use research. The results of the study provided useful information for the land-use administrative departments and land-users in Hebei Province. It helps to establish a dynamic and flexible mechanism of new construction land index allocation among regions. Additionally, this paper provided references for the implication of the "incremental-based" and "stock-based" planning. Adding time series and taking initial status of factors into consideration are suggested for further study.

**Author Contributions:** Conceptualization, M.L. and Y.S.; data curation: M.L. and W.D.; methodology, M.L. and A.C.; formal analysis, M.L. and N.W.; writing—original draft, M.L.; writing—review and editing M.L., Y.S. and J.H.; funding acquisition, J.H.

**Funding:** This work was supported by the National Key Technology Research and Development Program of the Ministry of Science and Technology of China (No. 2015BAD06B01).

**Conflicts of Interest:** The authors declare no conflict of interest.

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
