# Peer review of "Spatiotemporal Decoupling of Population, Economy and Construction Land Changes in Hebei Province"

_sustainability, doi:10.3390/su11236794_

Round 1
Reviewer 1 Report
I am attaching an annotated pdf with some comments.
In general, while the paper quality was good, I found it difficult to understand. It's pretty jargony, the various decoupling states. The "hot spot" maps also seemed unnecessarily abstract - to me it would have been clearer to simply map the variable itself (population or land or economy). I found myself thinking that the whole point of the paper, instead of using decoupling theory, could have been more easily and more clearly stated with a handful of maps and tables showing where those variables changed. But it might be that I am just too simplistic and this will be better appreciated by others.

Author Response
The authors would like to thank the reviewer for the serious review and comments, which are of great significance in guiding the manuscript. The manuscript was modified as suggested by the reviewer, and the responses to each suggestion are listed in attachment.
Please see the attachment.

Reviewer 2 Report
Interesting paper about the spatial relationships between Population, Economy and Construction Land Changes. The paper is relevant to the aims and scope of the journal.
I have a few considerations about the paper.
In the methods, the authors should give a more detailed description of the Hot Spots Analysis (Getis Ord Gi* Statistic) namely the relationship of population, economy and land construction changes when cold spots, cool spots, warm spots, and hot spots are defined.
Please revise the conclusion. The conclusions usually are not represented by a synthesis of the results. They should highlight for instance open research perspectives, so I suggest Authors report the conclusions to the general problem and how they have improved the scientific knowledge by carrying out this study.
In general, the figure legends are unreadable.
Author Response
The authors would like to thank the reviewer for the serious review and comments, which are of great significance in guiding the manuscript. The manuscript was modified as suggested by the reviewer, and the responses to each suggestion are listed in below.
Point 1: In the methods, the authors should give a more detailed description of the Hot Spots Analysis (Getis Ord Gi* Statistic) namely the relationship of population, economy and land construction changes when cold spots, cool spots, warm spots, and hot spots are defined.
Response 1: Accepted. The detailed description is added in section 2.3.3.
Point 2: Please revise the conclusion. The conclusions usually are not represented by a synthesis of the results. They should highlight for instance open research perspectives, so I suggest Authors report the conclusions to the general problem and how they have improved the scientific knowledge by carrying out this study.
Response 2: Accepted and revised in section 5.
Point 3: In general, the figure legends are unreadable.
Response 3: Accepted. Legends of figures are enlarged in this paper.